# Regioselective protein oxidative cleavage enabled by enzyme-like recognition of an inorganic metal oxo cluster ligand

Shorok A. M. Abdelhameed [1], Francisco de Azambuja [1], Tamara Vasović [2], Nada D. Savić [1], Tanja Ćirković Veličković[2,3,4,5] & Tatjana N. Parac-Vogt [1] ✉

Oxidative modifications of proteins are key to many applications in biotechnology. Metal-catalyzed oxidation reactions efficiently oxidize proteins but with low selectivity, and are highly dependent on the protein surface residues to direct the reaction. Herein, we demonstrate that discrete inorganic ligands such as polyoxometalates enable an efficient and selective protein oxidative cleavage. In the presence of ascorbate (1 mM), the Cu-substituted polyoxometalate $K_8[Cu^{2+}(H_2O)(\alpha_2\text{-}P_2W_{17}O_{61})]$, (Cu$^{II}$WD, 0.05 mM) selectively cleave hen egg white lysozyme under physiological conditions (pH =7.5, 37 °C) producing only four bands in the gel electropherogram (12.7, 11, 10, and 5 kDa). Liquid chromatography/mass spectrometry analysis reveals a regioselective cleavage in the vicinity of crystallographic Cu$^{II}$WD/lysozyme interaction sites. Mechanistically, polyoxometalate is critical to position the Cu at the protein surface and limit the generation of oxidative species to the proximity of binding sites. Ultimately, this study outlines the potential of discrete, designable metal oxo clusters as catalysts for the selective modification of proteins through radical mechanisms under non-denaturing conditions.

Protein oxidation, protein oxidative cleavage, and crosslinking[1,2] are oxidative modifications to the protein structure that have been linked to the onset of many diseases, including Alzheimer's disease and cancer[3,4]. Recent advances in the study of protein oxidative modifications (redox proteomics) streamlined the determination of the extent, and the location of these modifications[5]. Generally, the side chains of cysteine, methionine, tryptophan, tyrosine, and histidine were found to be the most prone to be oxidized, causing structural changes that may lead to the loss of protein three-dimensional structure and function[6]. Critically, to understand the influence of oxidation on the structure and function of proteins, oxidatively modified proteins need to be generated in a controlled manner. In the past, several chemical[7,8], electrochemical[9], and photochemical[10] techniques have been used, upgrading the mechanistic understanding, and opening the way for new potential treatments for several diseases[11,12]. However, these methods vary widely in their efficiency, selectivity, and degree of control over the reaction, as well as practicality and compatibility with physiological conditions. As a promising alternative, we introduce herein the potential of discrete metal-inorganic ligand complexes that can direct such oxidative modifications under mild non-denaturing conditions in a controllable fashion.

Metal-catalyzed oxidation (MCO) reactions are one of the most common methods to induce oxidative modification in proteins, given their straightforwardness to carry out, and compatibility with non-denaturing conditions, and different types of proteins; however, their efficacy is highly dependent on the coordination sphere of the metal[13–22]. In these reactions, redox-active metal ions/complexes (e.g., Cu$^{II}$, Co$^{II}$, Fe$^{II}$, Fe$^{III}$, and Ni$^{II}$), in the presence of a reducing agent such as ascorbic acid or an oxidant like O$_2$, generate reactive oxygen species (ROS) like HO$^•$ and O$_2^{•-}$ that end up oxidizing the protein[7,23]. However,

[1]KU Leuven, Department of Chemistry, Celestijnenlaan 200F, 3001 Leuven, Belgium. [2]Center of Excellence for Molecular Food Sciences & Department of Biochemistry, University of Belgrade - Faculty of Chemistry, Belgrade, Serbia. [3]Ghent University Global Campus, Yeonsu-gu, Incheon, South Korea. [4]Faculty of Bioscience Engineering, Ghent University, Ghent, Belgium. [5]Serbian Academy of Sciences and Arts, Belgrade, Serbia. ✉e-mail: tatjana.vogt@kuleuven.be

the reactivity of metal salts is highly dependent on the targeted protein containing strong surface exposed metal-binding sites[24–26]. Metal complexes featuring ligands that enable the redox-active unit to anchor onto specific residues of the protein have been shown a more fruitful strategy. In general, metal complexes generate ROS in the vicinity of the binding site, largely influencing the selectivity of oxidative modifications[27–29]. Remarkably, despite the progress observed in MCO methods for protein modification, the field is still largely dominated by metal complexes featuring organic ligands, which may also be oxidized by the ROS in solution causing loss of reactivity and selectivity[28–31]. In contrast, discrete inorganic ligands have been far less explored, being largely limited to heterogeneous systems whose selectivity control is yet rather limited[29,32]. In this context, we hypothesized that using an anionic polyoxometalate (POM) as the ligand to a redox-active Cu center could circumvent such stability drawbacks, while also imparting a high degree of selectivity under mild, controllable reaction conditions[33,34].

POM ligands exert exquisite control over the hydrolytic cleavage of proteins by favorably interacting with positive patches of the protein surface[35–41], but their directing ability has not been extended to other reaction modes yet. POMs are a diverse group of anionic clusters of oxygen and group IV–VI metals in their highest oxidation state, whose structural versatility renders their properties highly designable and useful[42–54]. In our previous work, we have extensively developed metal-substituted POM complexes (MS-POMs) as selective artificial proteases[33,40,41,55–60]. As such, MS-POMs selectively afford large peptide fragments in the mass range suitable for middle-down proteomics. Crystallographic and computational studies performed by us, and also others, have shown that POMs interact preferentially with positive patches of the protein surface, imparting high regioselectivity to the hydrolytic cleavage obtained[31–33,37,50,52,61]. Thus, we hypothesized the POM ligands could also be used to direct the selectivity in other types of reactions, such as the radical ones observed in MCO reactions[62–65]. Developing metal-POMs oxidative reactivity towards proteins would also circumvent key limitations of conventional metal complexes, as MS-POMs are stable to a wider range of pH[66], and the highly oxidized nature of the metals in their structure prevents competition with the protein substrate, increasing catalyst versatility, and life-span[66].

In this work, we set out to investigate the ability of a Cu[II]-substituted POM, the $K_8[Cu^{2+}(H_2O)(\alpha_2-P_2W_{17}O_{61})]$, (Cu[II]WD), to induce oxidative cleavage of a model protein substrate, hen egg white lysozyme (HEWL), in the presence of ascorbate (Asc) as a reducing agent and $O_2$. This detailed account highlights the inorganic POM ligand influence on the Cu[II] oxidative reactivity and selectivity, underlining the promising potential of such complexes for similar future applications.

## Results and discussion

To investigate the Cu[II]WD-mediated oxidative cleavage of HEWL, a small 129 amino acid protein (14.3 kDa) positively charged under physiological pH (isoelectric point 10.6–10.9)[67], we have initially incubated HEWL (0.02 mM) in the presence of Cu[II]WD (0.05 mM) and Asc (1 mM) at 37 °C and pH 7.5. After 1 h, the reaction was analyzed by a denaturing polyacrylamide gel in the presence of Tricine (SDS-PAGE) (Fig. 1). Under these conditions, the Cu[II]WD/Asc combination cleaved ~45% of HEWL producing four main bands with molecular weights of ~12.7, 11, 10 kDa, in addition to very faint band ~5 kDa. Following the reaction over 4 h showed no appearance of new bands, and no change in bands intensity, probably due to the full consumption of Asc within the first hour of the reaction (Supplementary Fig. 2). Moreover, the same bands were observed carrying out the reaction at pH 5.0, 7.5 or 8.5 (Supplementary Fig. 5), though the cleavage efficiency increased in the order pH 5.0 > 7.5 > 8.5. This trend is in agreement with a previous report, which has attributed the enhanced reactivity at pH ~4.5 to optimal production of ROS, mainly $H_2O_2$[68]. In addition to

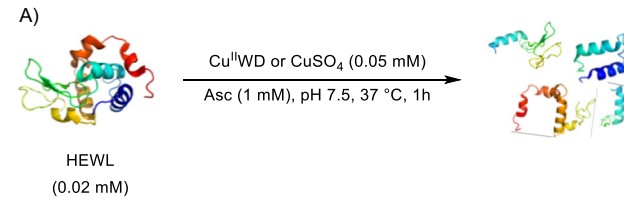

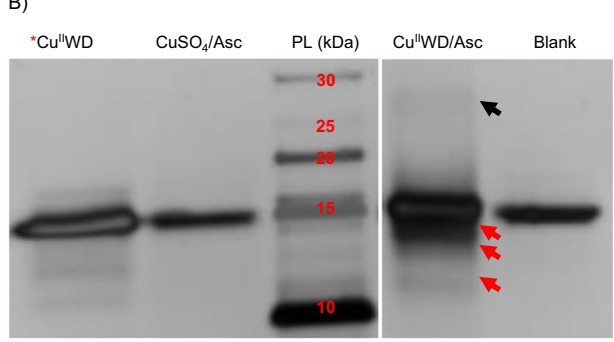

**Fig. 1 | Cleavage of HEWL catalyzed by Cu[II]WD/Asc. A** Equation shows the oxidative cleavage of HEWL (0.02 mM) in presence of Asc (1 mM) and Cu[II]WD or CuSO₄ (0.05 mM) at pH 7.5 and 37 °C for 1 h; **B** Silver stained SDS-PAGE gel of the oxidative cleavage of HEWL (0.02 mM) in presence of Asc (1 mM) and Cu[II]WD or CuSO₄ (0.05 mM) at pH 7.5 and 37 °C for 1 h. Red arrows indicate HEWL bands generated. The black arrow indicates the bands from HEWL dimerization. PL; protein ladder. *Hydrolytic cleavage of HEWL (0.02 mM) in presence of Cu[II]WD (2 mM) at pH 7.5 and 60 °C for 3 days. All experiments were repeated independently three times and provided similar results. Source data are provided as a Source Data file.

fragmentation, SDS-PAGE showed a band at larger MW of ~28 kDa, which was tentatively assigned to a potential HEWL-dimer, as metal-catalyzed protein oxidation is known to cause also protein oligomerization through covalent inter-crosslinking of surface exposed tyrosine or lysine residues[6,69,70].

In order to optimize HEWL cleavage, different Cu[II]WD/Asc ratios, as well as order of reagent addition were screened at pH 7.5 and 37 °C for 1 h. Firstly, HEWL (0.02 mM) cleavage was monitored using a constant concentration of Asc (1 mM), and different amounts of Cu[II]WD (0.005- 1 mM) (Fig. 2A and Supplementary Fig. 3). The highest yield of cleavage was observed when the concentration of Cu[II]WD was 0.05 mM. Increasing Cu[II]WD concentrations to ≥0.1 mM caused the formation of a precipitate, and less cleavage was observed. On the other hand, different concentrations of Asc (0.05–2 mM) were examined at a constant concentration of Cu[II]WD (0.05 mM) (Fig. 2B and Supplementary Fig. 4). Generally, the amount of cleaved HEWL directly correlated to the concentration of Asc. While at low Asc concentrations negligible cleavage was observed, ~70% of HEWL was cleaved upon increasing the Asc concentration to 2 mM Asc. Importantly, the same fragmentation pattern was observed regardless of the amount of Asc, underlining the high cleavage selectivity of Cu[II]WD-mediated oxidative cleavage. Finally, different orders of Cu[II]WD and Asc addition to HEWL did not affect the results, either regarding protein consumption, or fragmentation pattern.

### Mechanistic studies

Several mechanistic experiments based on schemes reported in the literature were conducted to gain a better understanding on the nature of the reaction[68,71]. In previously proposed mechanisms, Asc reduces the Cu[II] complex to Cu[I], which in turn is able to reduce an $O_2$ molecule from the air, leading to the formation of ROS. The ROS induces the formation of radicals on the protein, ultimately leading to protein cleavage (Fig. 3)[6,9,70,72]. Therefore, we carried out reactions in the absence of Asc, in the presence of radical inhibitors, and with

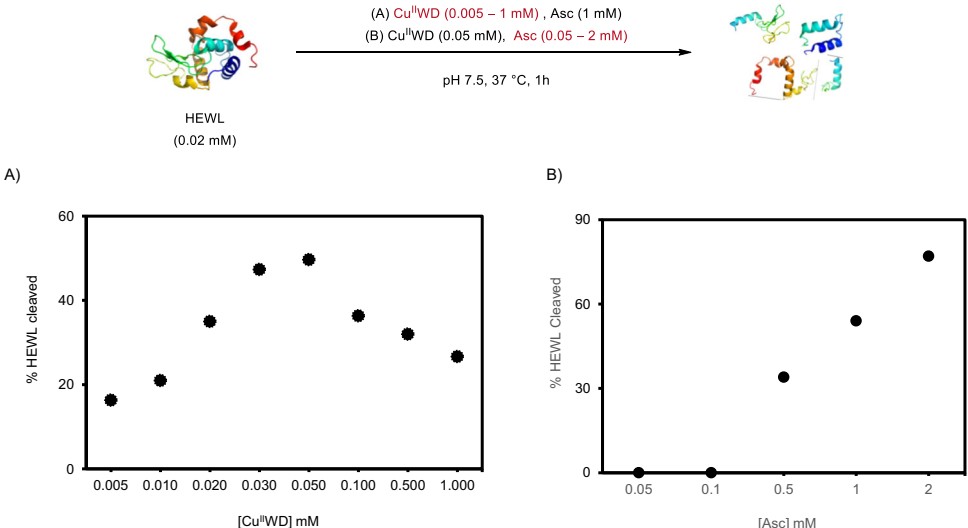

Fig. 2 | **Effect of Cu$^{II}$WD/Asc concentration on HEWL cleavage.** Amount of HEWL (%) cleaved in presence of: **A** Asc (1 mM) and different concentrations of Cu$^{II}$WD (0.005–1.00 mM); **B** Cu$^{II}$WD (0.05 mM) and different concentrations of Asc (0.05–2.00 mM), at pH 7.5 and 37 °C for 1 h. All experiments were measured two times (technical replicate) and provided similar results. Source data are provided as a Source Data file.

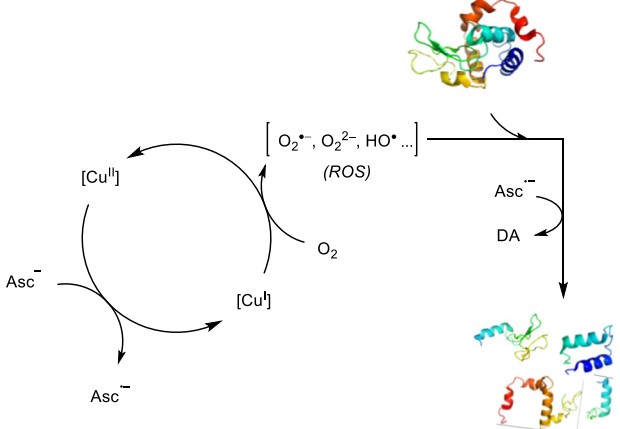

Fig. 3 | **Proposed mechanism of the reaction of Cu$^{II}$ with Asc.** Reactive oxygen species (ROS) are generated by Cu(I) intermediate reaction with molecular oxygen. Cu(I) intermediate is formed by oxidation of ascorbate in situ. DA = dehydroascorbic acid.

alternative oxidants, as well as electron paramagnetic resonance spectroscopy (EPR) experiments.

The mechanistic experiments evidenced the oxidative nature of the Cu$^{II}$WD mediated HEWL cleavage reaction in the presence of Asc (Tables 1, 2, and Fig. 4), (see full discussion Supplementary information section 2.2.). The non-hydrolytic nature of the reaction is supported by the different molecular weight of bands observed when purely hydrolytic and oxidative conditions were used (Fig. 1, Table 1, and Supplementary Fig. 14). Moreover, inhibition of the reaction in the presence of common radical inhibitors (Table 2, Supplementary Fig. 6 and 9), and the absence of reaction with the exclusion of oxygen (Supplementary Fig. 11) pointed to the involvement of radicals, which has been confirmed by the presence of characteristic sharp peaks of ascorbyl radical on the EPR spectra of Cu$^{II}$WD in the presence of Asc (Fig. 4). Subsequent formation of dehydroascorbic acid has also been observed (Supplementary Fig. 23). Intriguingly, no cleavage was detected when H$_2$O$_2$ was used instead of Asc (Supplementary Fig. 12), suggesting that any H$_2$O$_2$ eventually formed in solution as a consequence of oxygen reduction, was not responsible for the observed cleavage. This is in contrast with previous mechanistic proposals which suggested H$_2$O$_2$ as an intermediate in similar reactions[68], and implies that a mild oxidant species formed through the reaction of Cu$^{I}$WD and O$_2$ might be enabling HEWL oxidative cleavage[73,74].

## POM ligand enhances Cu reactivity and selectivity

The radical, and yet selective nature of the cleavage reaction led us to consider further the role of the POM ligand in the reactivity, especially regarding its potential influence on the cleavage selectivity observed (Fig. 1). To this end, control experiments comparing the reactivity, and protein binding of Cu$^{II}$WD with copper salts, and a detailed investigation by mass spectrometry of the sites where the protein sequence was cleaved were carried out.

## Control experiments

Control experiments evidenced that Cu$^{II}$WD affords superior reactivity compared to simple Cu salts, POM or POM/Cu salts combinations, and that POM binding to protein is necessary to trigger the observed reactivity. No cleavage was detected when simple Cu(II) salts like CuSO$_4$ and CuCl$_2$ (0.05 mM), or the lacunary Wells-Dawson POM (α$_2$WD, 0.05 mM) were used instead of Cu$^{II}$WD (Fig. 1 and Supplementary Fig. 1). In addition, a combination of CuSO$_4$ (0.05 mM)) and α$_2$WD (0.05 mM) was directly mixed with HEWL and Asc, or preincubated at 37 °C for 15 min, before the addition of HEWL and Asc. Only 15% of HEWL was cleaved in presence of CuSO$_4$/α$_2$WD in comparison to 45% observed in presence of Cu$^{II}$WD (0.05 mM) under the same conditions (Supplementary Fig. 1 and 10). These results showcase both the relevance of the POM ligand for the Cu-mediated reaction, and the advantage of using a pre-formed Cu$^{II}$WD complex. To further demonstrate that, the oxidative cleavage of HEWL was conducted in the presence of ethylenediaminetetraacetic acid (EDTA), a well-known strong chelating agent for Cu ions that remove the Cu ions from POM scaffold[75]. As a result, the cleavage was completely inhibited in the presence of EDTA (Supplementary Fig. 6). Monitoring the structure of Cu$^{II}$WD via $^{31}$P NMR spectroscopy in the presence and absence of EDTA revealed the disappearance of the Cu$^{II}$WD peak, and the appearance of two new peaks related to α$_2$WD (Supplementary Fig. 7). Thus, the low reactivity observed using CuSO$_4$/α$_2$WD combination probably arises from the equilibrium between the free Cu$^{II}$ ions and Cu$^{II}$-substituted α$_2$WD (Cu$^{II}$ + α$_2$WD ⇌ Cu$^{II}$WD), suggesting that only Cu$^{II}$-substituted WD actively cleaves the protein. These results contrast sharply with the absence of cleavage for proteins without surface exposed Cu-binding

**Table 1 | Hydrolytic and oxidative reaction conditions afford distinct cleavage of HEWL**

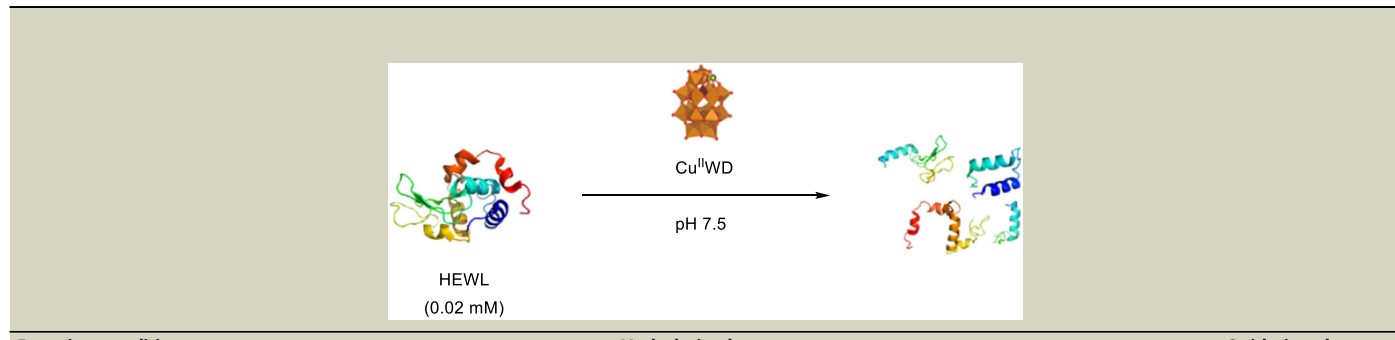

| Reaction conditions | Hydrolytic cleavage | Oxidative cleavage |
|---|---|---|
| [Cu$^{II}$WD] (mM) | 2 | 0.05 |
| [Asc] (mM) | - | 1 |
| T (°C) | 60 | 37 |
| MW of bands (kDa) | 11.5, 9.1, 7.6 | 12.7, 11, 10, 5 |

**Table 2 | Radical scavengers decrease the amount of HEWL cleaved by Cu$^{II}$WD/Asc**

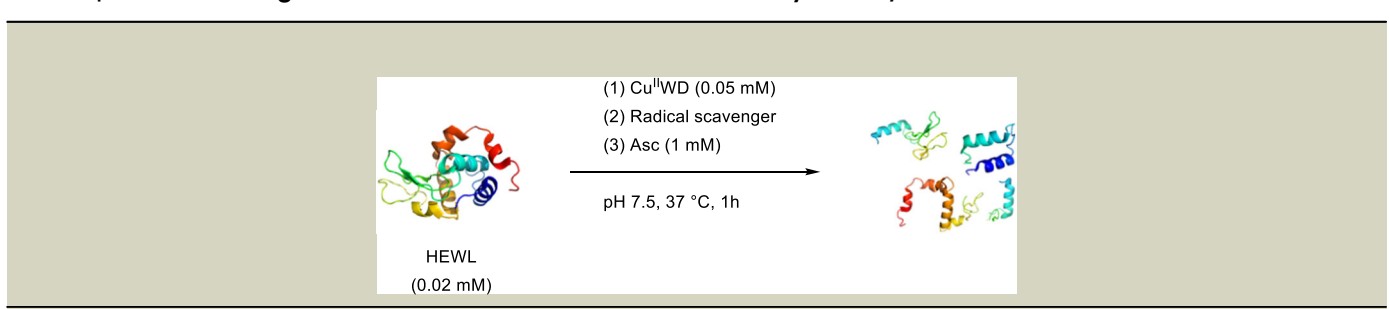

| Entry | Radical scavenger | Concentration (mM) | Cleavage (%) | Cleavage inhibition (%) |
|---|---|---|---|---|
| 1 | None | – | 45 | - |
| 2 | Mannitol[a] | 1.00 | 24 | 47 |
| 3 | Mannitol | 1.00 | 0 | 100 |
| 4 | GSH | 0.02 | 45 | - |
| 5 | GSH | 1.00 | 10 | 78 |
| 6 | Thiourea | 1.00 | 10 | 78 |
| 8 | N$_2$[b] | - | 0 | 100 |

[a]Mannitol was added directly after Asc.
[b]The reaction was conducted under N$_2$ atmosphere.

sites (mostly His) like HEWL[17], and the exclusive oxidation of only histidine-containing peptides observed previously[76], evidencing the crucial role of the POM ligand to trigger Cu oxidative reactivity. Such complexation-dependent reactivity is consistent with other Cu-mediated/catalyzed oxidative reactions[77,78]. Importantly, UV-Vis monitoring showed that the Cu$^{II}$WD structure remains unchanged in the absence and presence of HEWL and Asc under reaction conditions (Supplementary Fig. 18). Additionally, Cu$^{II}$WD showed high stability and integrity at 60 °C for 1 day at different pH conditions (Supplementary Fig. 16 and 17).

**Binding studies**

Tryptophan (Trp) fluorescence of HEWL in presence of CuSO$_4$, α$_2$WD, or Cu$^{II}$WD confirmed that the POM ligand interacts stronger with HEWL than Cu(II) ions, suggesting that it also promotes the Cu center approximation to the protein substrate. The quenching of Trp fluorescence is a straightforward tool to study protein interactions, and in order to probe whether the Cu or the POM drives Cu$^{II}$WD/HEWL interactions, the Trp fluorescence of HEWL in presence of increasing amounts of CuSO$_4$, α$_2$WD or Cu$^{II}$WD was monitored (Supplementary

Fig. 19). Clearly, α$_2$WD and Cu$^{II}$WD caused a much more pronounced quenching of HEWL fluorescence than CuSO$_4$. In addition, the virtually equivalent number of quencher molecules ($n$), and association constants ($K_a$) obtained for α$_2$WD or Cu$^{II}$WD, are markedly larger in comparison to CuSO$_4$, strongly suggesting that the POM scaffold is the one governing the interaction of Cu$^{II}$WD with the protein during the reaction (Table 3)[79]. Further on, $^1$H NMR spectroscopy and circular dichroism (CD) of HEWL in presence of Cu$^{II}$WD/Asc showed that most of the protein largely remains in its native form, (Supplementary Fig. 20 and 21), indicating that the POM ligand enables protein oxidative cleavage under non-denaturing conditions.

Furthermore, Cu$^{II}$WD/HEWL binding is required for the reaction to occur, as evidenced by the inhibition of cleavage observed in the presence of another metal-POM complex having affinity for the same binding sites[61,79]. More specifically, using a redox inert Zr-Keggin complex ([Zr(α-PW$_{11}$O$_{39}$]$^{-10}$, ZrK)[80], previously used for protein hydrolysis as a competitive binder resulted in a clear decrease of reaction efficiency. When ZrK (0.05 mM) was first mixed with Cu$^{II}$WD (0.05 mM) before Asc (1 mM) and HEWL (0.02 mM) were added, a 51% decrease in protein cleavage was observed by SDS-PAGE

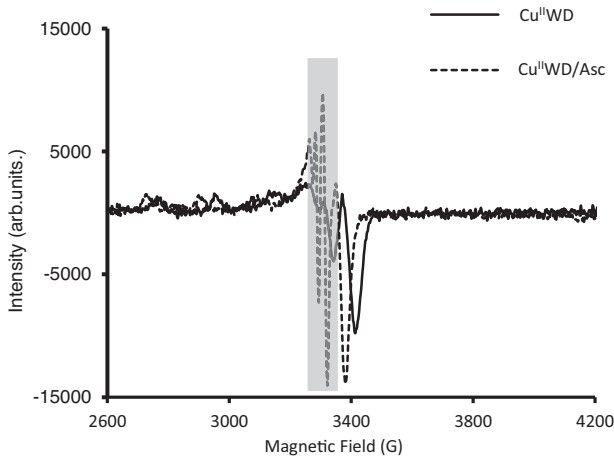

**Fig. 4 | EPR measurement of CuᴵᴵWD/Asc.** The appearance of sharp peaks (gray box) on the electron paramagnetic resonance spectroscopy (EPR) spectra of 0.05 mM CuᴵᴵWD in presence of 1.00 mM Asc indicates the presence of ascorbyl radical after the addition of Asc to CuᴵᴵWD. The results shown in Fig. 4 was measured repeatedly twice on same sample (technical replicate). Source data are provided as a Source Data file.

**Table 3 | Calculated values of the association constant ($K_a$, M⁻¹) and the number of the bound molecules of the quencher to the HEWL**

| Quencher | $n$ | $K_a$ (M⁻¹) |
|---|---|---|
| CuSO₄ | 0.61 | $1.82 \times 10^2$ |
| α₂WD | 1.62 | $6.91 \times 10^8$ |
| CuᴵᴵWD | 1.58 | $6.76 \times 10^8$ |

(Supplementary Fig. 8). Given that ZrK competes with CuᴵᴵWD for the same binding sites[32,71], the decrease observed strongly suggest that the binding of CuᴵᴵWD to HEWL is crucial for the reaction. Additionally, the oxidative cleavage reaction was conducted in presence of a high concentration of the Tris buffer solution (100 mM tris-HCl pH 7.5), as the Tris buffer has been previously reported to act as a shield around POMs[81]. The amount of cleavage observed by SDS-PAGE was negligible (Supplementary Fig. 13) in comparison to the cleavage activity at 10 mM tris-HCl pH 7.5. The high concentration of ions in solution likely disrupts the H-bonding and polar interactions between the negative shell of the POM ligand and the protein surface. These results evidence the POM role extends beyond being a simple ligand with a greater affinity for the protein surface, it actually resembles the nature of enzymes, whose reactivity is enabled only after an enzyme-substrate complex is formed in solution.

**Cleavage selectivity**

The overall enzyme-like nature of CuᴵᴵWD/Asc system prompted a more detailed investigation of the cleavage selectivity. To this end, SDS-PAGE bands were individually in-gel digested using trypsin, and analyzed via nano-scale liquid chromatography/tandem mass spectrometry (nLC-MS/MS)[82]. More specifically, we performed a comparative analysis of all semi- or non-tryptic peptides obtained from HEWL samples treated under identical conditions in the presence or absence of CuᴵᴵWD /Asc in order to identify the ones generated only by the action of CuᴵᴵWD/Asc (Fig. 5C and Supplementary Table 1). This protocol also allowed unambiguous assignment of which residues have been oxidized in presence of CuᴵᴵWD/Asc in the remaining intact protein (Table 4 and Fig. 5B).

The POM ligand largely governs the regioselectivity of the oxidative reactivity as evident from the detection of cleavage sites, and

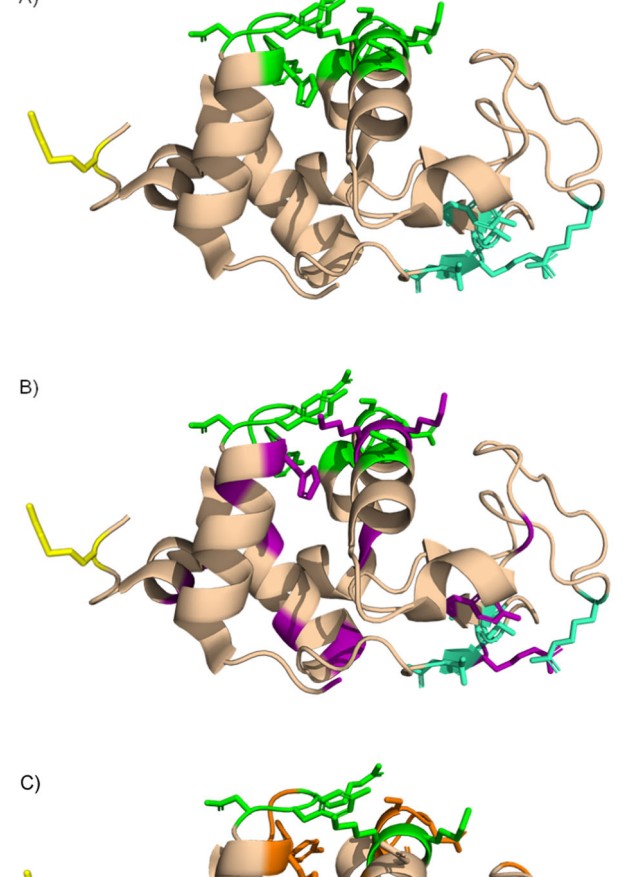

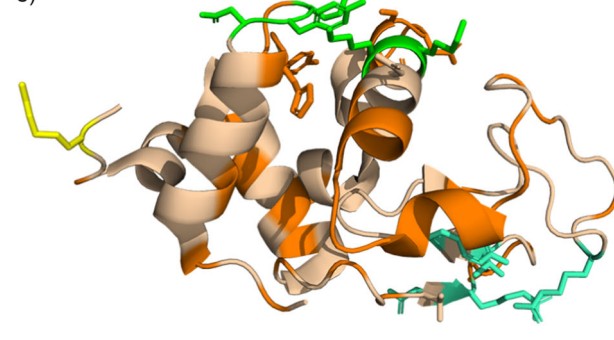

**Fig. 5 | Cartoon representation HEWL 3D structures. A** Cartoon representation of HEWL showing the three different crystallographic binding sites of CuᴵᴵWD (B1-[green]: His15, Gly16, Asn19, Tyr20, Arg21, Asn93, Lys96, and Lys97; B2-[green-cyan]: Asn44, Arg45, Asn46, Thr47; B3-[yellow]: Arg128, and the strongest MS-POM/HEWL interactions identified previously by molecular dynamics simulations[36,80]. **B** Cartoon representation of the main residues with side chain modifications (magenta) overlapped with CuᴵᴵWD/HEWL binding sites in the HEWL structure in the presence of CuᴵᴵWD/Asc. **C** Cartoon representation of the main regions that are oxidatively cleaved (orange) overlapped with CuᴵᴵWD/HEWL binding sites.

most of the oxidative modifications nearby previously reported binding sites (Fig. 5A). Precise assignment of the sites cleaved, and/or oxidatively modified in the presence of CuᴵᴵWD /Asc following the above protocol revealed that the CuᴵᴵWD /Asc system cleaves/modifies the protein in a highly regioselective fashion, meaning oxidation happens only over narrow regions of the protein sequence. More specifically, nLC-MS/MS results indicated that HEWL cleavage at His15-Gly16 and Gly16-Leu17 sites would produce bands around ~12.7 kDa, in agreement with the molecular weight of the bands observed in the SDS-PAGE analysis (Fig. 1). Similarly, the bands of ~11 kDa would originate from a cleavage between the residues Ser91 and Ser100, and cleavage in the region between the bonds Ala42-Thr43 and Ser50-Thr51 would generate the bands ~10 kDa. Bands with lower molecular weight ~5 kDa could be due to the cleavage at two or more sites at the

**Table 4 | Side-chain oxidative modifications found on the intact HEWL after reaction with Cu$^{II}$WD/Asc and the corresponding mass change**

| Residue | Side chain oxidative modification | Mass change (Da) |
|---|---|---|
| His15 | 2-Oxo-histidine | +16 |
| Trp28 | Indole ring hydroxylation, oxolactone | +16, +14 |
| Trp108 | Dihydroxy | +32 |
| Trp123 | Indole ring hydroxylation, Kynurenine, Oxolactone | +16, +4, +14 |
| Lys (1, 13, 33, 96, 97, 116) | Aminoadipic semialdehyde | −1.03 |
| Ser36 | Carbonyl formation | −2 |
| Arg45 | Deguanidination | −43 |
| Tyr53 | Dihydroxy | +32 |
| Pro79 | Oxidation to pyroglutamic acid | +14 |
| Thr118 | 2-Amino-3-oxo-butanoic acid | −2 |

same time. Strikingly, these segments of the sequence are all in the vicinity of Cu$^{II}$WD/HEWL interactions sites we have identified in a previous crystallographic study (Fig. 5). We have shown that Cu$^{II}$WD binds to HEWL on three different sites: the first (B1) involving residues His15, Gly16, Asn19, Tyr20, Arg21, Asn93, Lys96, and Lys97; the second (B2) consisting of Asn44, Arg45, Asn46, Thr47; and a third one (B3) involving only Arg128 (Supplementary Fig. 27)[79]. In addition, strong interactions of a MS-POM with Arg21, Thr43, Arg45 and Arg68 identified by molecular dynamics simulations have been proposed to direct the MS-POM to cleavage sites in its vicinity[35]. Together these data clearly indicate that the POM ligand drives the oxidative reactivity to relatively narrow intervals of the protein sequence due to its selective interactions with the protein surface, and the mild non-denaturing reactions conditions it enables to be used.

In conclusion, our results demonstrate that the enzyme-like recognition of the protein surface by a discrete metal oxo cluster inorganic ligand exerts a large influence in the efficiency and regioselectivity of a copper-mediated oxidative reaction. More specifically, using the Cu-substituted POM K$_8$[Cu$^{2+}$(H$_2$O)($\alpha_2$-P$_2$W$_{17}$O$_{61}$)] (Cu$^{II}$WD) in the presence of ascorbate (Asc), cleavage of the HEWL protein, and related oxidative modifications in the remaining intact protein, were observed in only four narrow regions of the protein sequence, all of which in the vicinity of crystallographic Cu$^{II}$WD/HEWL interaction sites. Moreover, unlike previously reported for other metal catalysts, the POM ligand sharply increases the catalyst affinity for the protein and dismisses the need for metal-binding sites on the protein surface to afford good reactivity. The enzyme-like recognition of the metal-oxo cluster is an important feature to enhance the selectivity and expand the scope of MCO reactions towards proteins in general. Finally, a detailed mechanistic investigation of the Cu$^{II}$WD/Asc system suggested that an oxidant species formed from the reaction of Cu$^I$WD and O$_2$ is likely the one responsible for the reactions observed under mild non-denaturing conditions. Overall, these findings not only highlight the redox-active metal-POM complexes as suitable catalysts for the oxidative modification or cleavage of proteins under mild non-denaturing conditions, but also underline the potential of well-defined designable metal oxo inorganic clusters as key ligands to direct different types of radical-based reactions in proteins.

## Methods
### Synthesis of $\alpha_2$-K$_{10}$P$_2$W$_{17}$O$_{61}$, ($\alpha_2$WD)
$\alpha_2$WD were prepared, according to previous literature report[83], as follow: 80 g (11.5 mmol) K$_6$[$\alpha/\beta$-P$_2$W$_{18}$O$_{62}$]·nH$_2$O was dissolved in 200 mL H$_2$O and 20 g (200 mmol) KHCO$_3$ in 200 mL H$_2$O was added. After 60 min of continuous stirring at room temperature, a white precipitate was collected on a glass filter and left to air dry. The final product, K$_{10}$[$\alpha_2$-P$_2$W$_{17}$O$_{61}$]·20H$_2$O, ($\alpha_2$WD), was characterized by $^{31}$P NMR: δ = −7.2 and −14.4 ppm.

### Synthesis of K$_8$[Cu$^{2+}$(H$_2$O)($\alpha_2$-P$_2$W$_{17}$O$_{61}$)], (Cu$^{II}$WD)
Cu$^{II}$WD were prepared, according to previous literature report[83], as follow: 10.0 g (2.07 mmol) of lacunary K$_{10}$[$\alpha_2$-P$_2$W$_{17}$O$_{61}$]·20 H$_2$O was dissolved in 20 ml H$_2$O (90 °C) under vigorous stirring. Then, 0.60 g (2.40 mmol) CuSO$_4$· 5H$_2$O in 8 ml H$_2$O was added instead was added and the solution was stirred for 15 min (90 °C). After cooling at 4 °C overnight, light-green crystals formed, which were collected on a glass filter and recrystallized once in 10 mL H$_2$O (90 °C). The K$_8$[Cu$^{2+}$(H$_2$O)($\alpha_2$-P$_2$W$_{17}$O$_{61}$)], (Cu$^{II}$WD). crystals were collected after cooling at 4 °C overnight and left to air dry at room temperature. Resulting green crystals were validated by $^{31}$P NMR: δ = −13.0 ppm.

### Oxidative cleavage of HEWL
Solutions containing HEWL (0.02 mM), Cu$^{II}$WD or Cu salts (CuCl$_2$, CuSO$_4$) (0.05 mM), and Asc or H$_2$O$_2$ (1 mM) were prepared in 10.0 mM tris-HCl buffer (pH 7.5). Samples were incubated at 37 °C and aliquots were taken at different time increments and analyzed by SDS-PAGE.

### Hydrolytic cleavage of HEWL
Solutions containing HEWL (0.02 mM), Cu$^{II}$WD (2 mM) were prepared in 10.0 mM tris-HCl buffer (pH 7.5). Samples were incubated at 60 °C for 3 days. Aliquots were taken and analyzed by SDS-PAGE.

### Electron paramagnetic resonance
A 1.5 mL centrifuge tube was charged with 5 μL of 10 mM Cu$^{II}$WD stock solution, 5 μL of 200 mM Asc stock solution and 20 μL of 1 mM of HEWL stock solution, and enough 10.0 mM tris-HCl (pH 7.5) buffer was added to complete the reaction volume to a total of 1 mL (final concentrations of Cu$^{II}$WD, Asc and HEWL were 0.05, 1 and 0.02 mM, respectively). The reaction mixture was homogenized using a vortex. Next, 300 μL of the reaction mixture were transferred to an EPR tube. Immediately afterwards, the tube was immersed in liquid N$_2$. EPR measurements were conducted at 160 K on a X-band Bruker 300E continuous wave spectrometer with a rectangular cavity.

### Fluorescence spectroscopy
Tryptophan fluorescence quenching studies of HEWL (10.0 μM) solution (10.0 mM tris-HCl buffer, pH 7.5) at room temperature. The emission was monitored from 300 nm to 420 nm. Sample was irradiated with a wavelength of 295 nm to avoid excitation of tyrosine residues, with a maximum at ~340 nm. Fluorescence spectra were measured with increasing concentrations of Cu$^{II}$WD, $\alpha_2$WD, or CuSO$_4$ (0 to 15.0 μM). Emission fluorescence spectra were recorded on a FS-920P spectrofluorimeter (Edinburgh Instruments, Livingston, UK). Quartz cuvettes with 10.0 mm optical path length were used.

### Reporting summary
Further information on research design is available in the Nature Portfolio Reporting Summary linked to this article.

## Data availability
The authors declare that the data supporting the findings of this study are available within the article and its Supplementary Information file; source data, including the uncropped SDS-PAGE gels, is provided with the paper as Source Data file. The mass spectrometry proteomics data generated in this study have been deposited to the ProteomeXchange Consortium via the PRIDE partner repository with the dataset identifier PXD037983. Source data are provided with this paper.

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

## Acknowledgements

We thank KU Leuven and the Research Foundation—Flanders (FWO), the Ministry of Education, Science and Technological Development of the Republic of Serbia [contract no. 451-03-9/2021-14/200168], the Serbian Academy of Sciences and Arts [project no. F-26], and the European Union's Horizon 2020 research and innovation program under project FoodEnTwin (grant agreement no. 810752) for funding. S.A.M.A. (1115321N), F.d.A. (195931/1281921N), N.D.S. (1267623N) thank the FWO for fellowships. We thank professor Dirk De Vos and Dr. Simon Smolders for helping with the EPR measurements.

## Author contributions

Conceptualization, T.P.V., F.d.A., S.A.M.A.; experiments: S.A.M.A., N.D.S. (revision); data analysis and interpretation: S.A.M.A., F.d.A.; mass spectrometry (data acquisition, analysis, and interpretation): S.A.M.A., T.V., T.C.V., writing—original draft, S.A.M.A., F.d.A.; writing—review & editing, S.A.M.A., F.d.A., T.V., T.C.V., T.P.V.; supervision, T.P.V.

## Competing interests

The authors declare no competing interests.
