## [Peer Review File · Nature Communications]

Regioselective protein oxidative cleavage enabled by enzyme-like recognition of an inorganic metal oxo cluster ligandREVIEWER COMMENTS

Reviewer #2 (Remarks to the Author):

The author's purpose of the topic entitled "All-Inorganic Ligand Featuring an Enzyme-Like Recognition Enables Rapid and Regioselective Protein Cleavage Under Oxidative Conditions" is interesting, also for scientists from related research fields. I would recommend the suggestions described below:

- 1) The title should be short and concise. That would favor future citations. What is new in the paper?
- 2) The abs should and/or could be, at least in part, a mirror of the paper. Abstract should be also quantitative as possible for rapid comparison with others studies, referring for instance POMs concentration used.
- 3) Recent references could be updated from 2021 and 2022 (only 4 in the paper) for metals and POMs interaction with proteins and biomedical applications, for example from Garrriba et al 2021 (CCR), Aureliano et al, 2021, 2022 (CCR), Scibior 2021,2022, Gumerova et al, 2018 (For POTs), among others, once one of the topic is the putative interaction of metals and POMs with proteins and also inhibitors of enzymes.
- 4) What is the pH vale of the POMs stock solutions used?
- 5) Did the authors check for the stability of the P2W17 compound used the concentration used and at the experimental conditions?

Reviewer #4 (Remarks to the Author):

The authors report a new approach to use purely inorganic, Cu-functionalized polyoxometalates (POMs) as selective and efficient catalysts for oxidative cleavage of proteins at selected residues only.

This is an original and highly important study, as it demonstrates how the catalytic activity of bio-compatible inorganic metal oxide anions can be used for regioselective protein cleavage under physiological conditions. This could break new ground for the development of in vitro and in vivo applications in the fields of bio-sensorics and theranostics.

When comparing to the state of the art, this report is a significant step forwards for the reasons outlined above, making it principally suitable for publication in Nature Communications.

There are a few comments for consideration by the authors which might help to shed further light on the underlying reaction mechanism:

1. Have the authors performed reference experiments where CuWD and Asc are reacted under the given conditions and the (possible) changes in UV-Vis-NIR are monitored over time? This could be done in the presence and absence of oxygen. Purging of the solution with oxygen while further monitoring the spectral changes might provide valuable understanding of the rates (and rate-limiting processes in their system (e.g. oxygen transfer from gas to liquid, see here for example: <https://pubs.rsc.org/en/content/articlelanding/2016/dt/c6dt03051c>).
2. One additional control experiment which the authors could consider (to demonstrate the role of the intact Cu-functionalized WD): the "standard" cleavage reaction could be performed in the presence of a strong Cu(II) complexing agent, eg EDTA. This should rip out the Cu(II) from the POM, and in turn should disable the reactivity. This approach has been used before to explore the role of Cu(II) coordination to oxidative POM catalysts, see here: <https://doi.org/10.1002/slct.201701321>

In the SI:

1. Can the authors state which analyses were used to assess the purity of the literature-known materials? This is slightly vague at the moment.
2. Can the authors use the correct formulae nomenclature (square bracket notation) for the POMs, please?

Overall, this manuscript is well written, innovative and convincing in its methodologies and data reported. Once the authors have addressed the comments, I expect that publication can be recommended.

Reviewer #5 (Remarks to the Author):

The authors present a copper substituted Wells Dawson polyoxometalate (CuIIDW) for the oxidative cleavage of hen egg white lysozyme (HEWL) protein. The work is very well conducted and the authors report solid evidences for the mechanism of the reaction in air using ascorbate as reducing agent. Different to other catalysts used for the oxidative cleavage of proteins, this system cleaves HEWL in defined regions which correlate well with the interaction sites of HEWL with CuIIDW. Further, the protein is not denatured under the reaction conditions. The selectivity, stems from the interaction of CuIIDW with defined sites of HEWL, which correspond mainly to positively charged regions on the surface of the protein, as expected from a polyanion as CuIIDW. This interaction is easier with a protein as HEWL, that is positively charged at the reaction conditions. It would have been interesting to have another example with a protein with a much lower isoelectric point.

Nevertheless, in my opinion the novelty and selectivity of the system compared to the state of the art justify the publication of this work after taking into consideration the comments below:

1.- In line 59 the authors state: "Remarkably, despite the progress observed in metal-catalysed oxidations (MCO) methods for protein modification, the field is still largely dominated by metal complexes featuring organic ligands, which may also be oxidized by the ROS in solution causing loss of reactivity and selectivity."

Can the authors compare their system with MCO methods regarding this issue? Have they tried to recycle the catalyst or by using spaced discrete additions of HEWL to the CuIIDW/Asc mixture and monitoring the SDS-PAGE?

2.- The control under purely hydrolytic conditions is used repeatedly along the work, as appropriate. However, this leads to certain inconsistencies in the work because the results do not seem to be reproducible. For instance, in Table 1 the reported MW are 12.7, 11, 10 and 5 kDa, however those bands cannot be seen in the corresponding lane in Figure 1. Figure S10, reports the same experiment with a much better resolution. The bands shown in this Figure S10, and reported next to it are not the same as in the main text. I think the authors should clarify this issue. Plus, I would ask they used the conditions for the gel in Figure S10 where possible as the resolution in the other gels is very low.

3.- In the supporting information the authors state: "Figure S10, unambiguously, reveals that the bands produced from the cleavage of HEWL via CuIIDW is different from CuIIDW /Asc". Figure S10 is a SDS-PAGE gel for the purely hydrolytic conditions, but the resolution is much higher than for the others. Both reactions should be compared with SDS-PAGE under the same high-resolution conditions.

4.- In Figure 2, the HEWL:CuIIDW ratios are screened with [HEWL]=0.02 mM. Taking into account that there are three interaction sites in HEWL. Have the authors checked also more concentrations in the range 0.05 mM-1.0 mM?

5.- In Table 2 (Entry 2), how long after Asc was the mannitol added?

6.- "In lines 169 to 171: "The low reactivity observed using a combination of the CuSO₄/α2WD combination probably arises from the equilibrium between the free CuII ions and CuII-substituted α2WD (CuII + α2WD ⇌ CuIIDW), suggesting that only CuII substituted WD actively cleaves the protein." It would be interesting to check this with two experiments: 1) Preincubation of the lacunary WD and the CuII salt before setting the reaction with Asc and HEWL, and 2) preincubation CuIIDW before setting the reaction with HEWL and Asc.

7.- lines 200-202: "These results evidence the POM role extends beyond being a simple ligand with a greater affinity for the protein surface, it actually resembles the nature of enzymes, whose reactivity is enabled only after an enzyme substrate complex is formed in solution." In my opinion, both parts of the sentence have very similar meanings, but if I understand well the difference, I don't think the authors have really that evidence. Maybe a control experiment with high ionic strength medium to screen ionic interactions could provide insight into that. That would also help defining the nature of the recognition interactions beyond the crystal structure.

8.- Line 248: which are the four narrow regions for the modification? In the context of oxidative cleavage and binding it was limited to three regions. Could the authors clarify that?

9.- Concerning other techniques. In the supporting information CD results are discussed. The authors report minima "at $\lambda = 208$ nm and a smaller minimum at $\lambda = 222$ nm, both characteristic for α -helical structure elements. The minimum at $\lambda = 215$ nm, characteristic of β -sheet elements..." I am not able to distinguish the three minima in Figure S16, specially the one at 215 nm. It would be helpful if the authors indicate them in the graph.

Finally, I have detected these typos and mistakes in references to figures:

L84: Asc abbreviation for ascorbate has not been introduced yet.

L141: It should be contrast instead of contrasts.

L212 and L221: I think it should be Figure S21 in place of S19. Figure 4 would be even better as Figure S21 is already included in Figure 4.

L224: Figure S18 is an NMR spectrum. Maybe Figure S21 or Figure 4 is more adequate here.

L245: it should be copper-mediated instead of copper-mediate.

Possible mistakes in the supporting information:

a) In the section "Preparation of known compounds", it should be a instead of "a" in several compounds.

b) In the circular dichroism section there are several places where it should be λ instead of l.

c) Table S1, entry 79: should it be purple?

A. Response to reviewer's comments

Reviewer #2:

1. *The title should be short and concise. That would favor future citations. What is new in the paper?*

Response: After careful evaluation, we have updated our title to “Regioselective Protein Oxidative Cleavage Enabled by an Enzyme-like Recognition of an Inorganic Metal Oxo Cluster Ligand”.

2. *The abs should and/or could be, a least in part, a mirror of the paper. Abstract should be also quantitative as possible for rapid comparison with others studies, referring for instance POMs concentration used.*

Response: We have revisited our abstract to comply with journal's guidelines, and taking into consideration reviewer's suggestion. The abstract now reads:

“Oxidative modifications of proteins are key to many applications in biotechnology. Metal-catalyzed oxidation reactions (MCO) efficiently oxidize proteins but with low selectivity, and are highly dependent on the protein surface residues to direct the reaction. Herein, we demonstrate that discrete inorganic ligands such as polyoxometalates (POMs) enable an efficient and selective protein cleavage. In the presence of ascorbate (Asc, 1 mM), the Cu-substituted POM $K_8[Cu^{2+}(H_2O)(\alpha_2-P_2W_{17}O_{61})]$, ($Cu^{II}WD$, 0.05 mM) selectively cleaved hen egg white lysozyme (HEWL) under physiological conditions (pH =7.4, 37 °C) producing only four bands in the gel electropherogram (12.7, 11, 10, and 5 kDa). Liquid chromatography/mass spectrometry analysis revealed a regioselective cleavage in the vicinity of crystallographic $Cu^{II}WD/HEWL$ interaction sites. Mechanistically, the POM is critical to position the Cu at the protein surface, and limit the generation of oxidative species to the proximity of binding sites. Ultimately, this study outlines the potential of discrete, designable metal oxo clusters as catalysts for the selective modification of proteins through radical mechanisms under non-denaturing conditions.”

3. *Recent references could be updated from 2021 and 2022 (only 4 in the paper) for metals and POMs interaction with proteins and biomedical applications, for example from Garrriba et al 2021 (CCR), Aureliano et al, 2021, 2022 (CCR), Scibior 2021,2022, Gumerova et al, 2018 (For POTs), among others, once one of the topic is the putative interaction of metals and POMs with proteins and also inhibitors of enzymes.*

Response: We thank the reviewer for drawing our attention to this point. We have update our references. Following reviewer's suggestion relevant recent works, including those suggested, are numbered as 48-52. For convenience, the references included are listed below:

51. Gumerova, N. I.; Al-Sayed, E.; Krivosudský, L.; Ćipčić-Paljetak, H.; Verbanac, D.; Rompel, A. Antibacterial Activity of Polyoxometalates against *Moraxella Catarrhalis*. *Front Chem* **2018**, *6* (AUG). <https://doi.org/10.3389/fchem.2018.00336>.
52. Aureliano, M.; Gumerova, N. I.; Sciortino, G.; Garribba, E.; McLauchlan, C. C.; Rompel, A.; Crans, D. C. Polyoxidovanadates' Interactions with Proteins: An Overview. *Coordination Chemistry Reviews*. Elsevier B.V. March 1, **2022**. <https://doi.org/10.1016/j.ccr.2021.214344>.
53. Aureliano, M.; Gumerova, N. I.; Sciortino, G.; Garribba, E.; Rompel, A.; Crans, D. C. Polyoxovanadates with Emerging Biomedical Activities. *Coordination Chemistry Reviews*. Elsevier B.V. November 15, 2021. <https://doi.org/10.1016/j.ccr.2021.214143>.
54. Fabbian, S.; Giachin, G.; Bellanda, M.; Borgo, C.; Ruzzene, M.; Spuri, G.; Campofelice, A.; Veneziano, L.; Bonchio, M.; Carraro, M.; Battistutta, R. Mechanism of CK2 Inhibition by a Ruthenium-Based Polyoxometalate. *Front Mol Biosci* **2022**, *9*. <https://doi.org/10.3389/fmolb.2022.906390>.

55. Chang, D.; Li, Y.; Chen, Y.; Wang, X.; Zang, D.; Liu, T. Polyoxometalate-Based Nanocomposites for Antitumor and Antibacterial Applications. *Nanoscale Adv* **2022**. <https://doi.org/10.1039/D2NA00391K>.

4. *What is the pH value of the POMs stock solutions used?*

Response: The pH of POMs' stock solution is 7.5. We have indicated this experimental detail in the section "General Remarks" of the Supplementary Information.

5. *Did the authors check for the stability of the P_2W_{17} compound used [at] the concentration used and at the experimental conditions?*

Response: We have showed that $Cu^{II}WD$ is stable under the reaction conditions by comparing the integration of the peak at (-13 ppm) in the ^{31}P NMR spectra before and after the reaction (**Supplementary Fig. 16**). In order to obtain a good signal/noise ratio in this case, a concentration of $Cu^{II}WD$ higher (2 mM) than the one used in the oxidative cleavage reaction (0.05 mM) was used to circumvent the low sensitivity due to the paramagnetic effect of the copper. To complement this data, we have now included the UV-Vis spectra of various solutions containing $Cu^{II}WD$ in the same concentration (0.05 mM) used for the oxidative cleavage of HEWL. As it can be seen in the **Supplementary Fig. 18** of the revised Supplementary Information, the UV-Vis spectra of $Cu^{II}WD$ are the same before, and after the reaction. The increase in absorption intensity is due to the absorption of Asc ($\lambda_{max} = 266$ nm) and HEWL ($\lambda_{max} = 220$ and 280 nm).

Reviewer #4:

1. *Have the authors performed reference experiments where $CuWD$ and Asc are reacted under the given conditions and the (possible) changes in UV-Vis-NIR are monitored over time? This could be done in the presence and absence of oxygen. Purging of the solution with oxygen while further monitoring the spectral changes might provide valuable understanding of the rates (and rate-limiting processes in their system (e.g. oxygen transfer from gas to liquid, see here for example: <https://pubs.rsc.org/en/content/articlelanding/2016/dt/c6dt03051c>).*

Response: We thank the reviewer for this thoughtful comment. We agree that a deeper mechanistic investigation is crucial to take full advantage of our findings. Our attempts to monitor the reaction by UV-Vis spectroscopy have shown that the λ_{max} of ascorbate ($\lambda_{max} = 265$ nm), and its oxidation product dehydroascorbic acid ($\lambda_{max} = 185$ nm) overlap with the $Cu^{II}WD$ spectrum at pH 7 (seen in the **Supplementary Fig. 18** of the revised Supplementary Information), thus making UV-Vis not particularly useful technique to measure the rates of reaction. Further, the UV-Vis spectrum of $Cu^{II}WD/HEWL$ shows no clear changes in presence of Asc, most likely because the main band (at ca 280 nm) originates from the POM scaffold which should not be affected by the addition of Asc. We believe that the mechanistic studies presented so far are sufficient to support the main conclusions of our current work, and plan to address full mechanistic details in a separate future study.

2. *One additional control experiment which the authors could consider (to demonstrate the role of the intact Cu -functionalized WD): the "standard" cleavage reaction could be performed in the presence of a strong $Cu(II)$ complexing agent, eg EDTA. This should rip out the $Cu(II)$ from the POM, and in turn should disable the reactivity. This approach has been used before to explore the role of $Cu(II)$ coordination to oxidative POM catalysts, see here: <https://doi.org/10.1002/slct.201701321>*

Response: We thank the reviewer for this comment. The control experiment suggested was already included in **Supplementary Fig. 6** of the Supporting Information, and as predicted by the reviewer, no oxidative cleavage is observed in the presence of EDTA. To show this absence of cleavage is related to the absence of Cu^{II}WD in solution, an additional control experiment was done by incubating EDTA (2mM) with Cu^{II}WD (2 mM) at pH 7.5 and 37 °C for 1h. The change in the color of the solution from colorless to pale blue indicated the formation of Cu^{II}-EDTA complex (*J. Phys. Chem. B* **1997**, *101*, 1857). Additionally, the structure of Cu^{II}WD was monitored via ³¹P NMR spectroscopy in the presence and absence of EDTA. **Supplementary Fig. 7** shows of ³¹P NMR spectra of Cu^{II}WD in the presence and absence of EDTA. In presence of EDTA, the disappearance of peak related to Cu^{II}WD and the appearance of two other new peaks. To check if the two new peaks related to α_2 WD, the sample was spiked with α_2 WD. Only two peaks appeared in this new analysis. Although slight changed in chemical shift, these single pair of peaks indicates that the new peaks observed in the presence of EDTA are indeed related to α_2 WD. To clarify this key mechanistic information, we have included these results in the paragraph about ‘Control experiments’ in the section “POM ligand enhances Cu reactivity and selectivity”, and the relevant experimental details have been added, **Supplementary Fig. 6, and 7** in section (2.1. oxidative cleavage of the Supplementary Information).

In the SI:

1. *Can the authors state which analyses were used to assess the purity of the literature-known materials? This is slightly vague at the moment.*

Response: We have revised the part “Preparation of known compounds” in section 1 of our Supplementary Information to include that the identity and purity of the literature-known materials were evaluated via using ³¹P NMR spectroscopy, as previously reported in the literature.

2. *Can the authors use the correct formulae nomenclature (square bracket notation) for the POMs, please?*

Response: The formula form (α_2 -K₈P₂W₁₇O₆₁(Cu²⁺.OH₂)) has been amended to the correct formula nomenclature (K₈[Cu²⁺(H₂O)(α_2 -P₂W₁₇O₆₁)]).

Reviewer #5:

1. *In line 59 the authors state: “Remarkably, despite the progress observed in metal-catalysed oxidations (MCO) methods for protein modification, the field is still largely dominated by metal complexes featuring organic ligands, which may also be oxidized by the ROS in solution causing loss of reactivity and selectivity.” Can the authors compare their system with MCO methods regarding this issue? Have they tried to recycle the catalyst or by using spaced discrete additions of HEWL to the Cu^{II}WD/Asc mixture and monitoring the SDS-PAGE?*

Response: We thank the reviewer for this question. Since there is not any reported study about oxidative cleavage of HEWL by metal complexes, the direct comparison with our system is not possible. In addition, as discussed in our answers to Reviewer 4, the Cu^{II}WD catalyst is stable under the reaction conditions used. We have demonstrated this by using both ³¹P NMR, and UV-Vis spectroscopy (**Supplementary Fig. 16 - 18** of the revised Supplementary Information). Thus, it is reasonable to suppose the catalyst recyclability would be feasible. However, the amount of catalyst used in our test reactions is rather small to attempt any meaningful recyclability experiment. Moreover, in our experiments the protein is only partially converted, making monitoring the reactions after spaced discrete additions of HEWL a complex task. Finally, the literature available extensively support our statements. Previous

reports already demonstrated that metal complexes with organic ligands have a high tendency to be attacked by the radical produced in the system which cause the loss of their catalyst activity (*Journal of Water Process Engineering* 2020, 36, 101320; *Photochemistry and photobiology* 1978, 28, 681), and that POMs showed high structural stability in the presence of radicals, especially the Wells-Dawson type POMs (Activation of Hydrogen Peroxide by Polyoxometalates. In *Mechanisms in Homogeneous and Heterogeneous Epoxidation Catalysis*, 2008, pp. 155-176. DOI: 10.1016/B978-0-444-53188-9.00004-3; *Journal of Molecular Catalysis A: Chemical* 2007, 262, 67). Therefore, although we agree with the reviewer that a comparison between classical MCO catalysts and our system is worth studying, we believe this point is beyond our present work. To address the point raised by the referee, we have included the references indicated above in our introduction.

2. *The control under purely hydrolytic conditions is used repeatedly along the work, as appropriate. However, this leads to certain inconsistencies in the work because the results do not seem to be reproducible. For instance, in Table 1 the reported MW are 12.7, 11, 10 and 5 kDa, however those bands cannot be seen in the corresponding lane in Figure 1. Figure S10, reports the same experiment with a much better resolution. The bands shown in this Figure S10, and reported next to it are not the same as in the main text. I think the authors should clarify this issue. Plus, I would ask they used the conditions for the gel in Figure S10 where possible as the resolution in the other gels is very low.*

Response: We are appreciative of reviewer's careful critical reading. It is important to note that we have repeated the cleavage experiments many times throughout our work, and all of them were analyzed employing the same SDS-PAGE protocol which we have been using successfully for the last 10 years. The results across several experiments were always consistent. Thus, we are very confident the results are reproducible. We believe reviewer's confusion might have arisen from an unfortunate clerical mistake in **Table 1**. The fragments with MW 12.7, 11, 10 and 5 kDa should have been under the oxidative cleavage while the fragments with molecular weight 11.5, 9.1, 7.6 kDa were observed in the cleavage conducted under hydrolytic conditions. Table 1 has been revised accordingly.

3. *In the supporting information the authors state: "Figure S10, unambiguously, reveals that the bands produced from the cleavage of HEWL via Cu^{II}WD is different from Cu^{II}WD /Asc". Figure S10 is a SDS-PAGE gel for the purely hydrolytic conditions, but the resolution is much higher than for the others. Both reactions should be compared with SDS-PAGE under the same high-resolution conditions.*

Response: As mentioned in the previous answer, all cleavage experiments were analyzed using the same SDS-PAGE protocol. However, after cleavage reaction the solutions containing POMs and proteins eventually form a precipitate that could interfere with quality of the bands in the gel. For this reason, we repeated the analysis exhaustively to make sure the results are real and reproducible. Thus, we are once more confident that the oxidative and hydrolytic conditions afford fragments with different molecular weights. To clarify this in the Supplementary Information, we have revised **Supplementary Fig. 14** (former Figure S10) to include an SDS-PAGE electropherogram in which oxidative and hydrolytic cleavage reactions were analyzed together, and adjusted the corresponding discussion presented in the same section. As seen in the **Supplementary Fig. 14**, the fragmentation pattern for each is clearly different from the other.

4. *In Figure 2, the HEWL:Cu^{II}WD ratios are screened with [HEWL] = 0.02 mM. Taking into account that there are three interaction sites in HEWL. Have the authors checked also more concentrations in the range 0.05 mM - 1.0 mM?*

Response: We have tried the same experiments using 0.04 mM of HEWL. In general, the same results were observed. However, increasing the concentration of HEWL > 0.04 mM in presence of Cu^{II}WD led to more frequent formation of a precipitate (likely involving the protein). Thus, we opted to work at lower HEWL concentrations to ensure consistency among experiments.

5. In Table 2 (Entry 2), how long after Asc was the mannitol added?

Response: The mannitol was added directly after the addition of Asc. To clarify this aspect, we inserted the word “directly” in the footnote of Table 1.

6. “In lines 169 to 171: “The low reactivity observed using a combination of the CuSO₄/α₂WD combination probably arises from the equilibrium between the free Cu^{II} ions and Cu^{II}-substituted α₂WD (Cu^{II} + α₂WD ⇌ Cu^{II}WD), suggesting that only Cu^{II} substituted WD actively cleaves the protein.” It would be interesting to check this with two experiments: 1) Preincubation of the lacunary WD and the Cu^{II} salt before setting the reaction with Asc and HEWL, and 2) preincubation Cu^{II}WD before setting the reaction with HEWL and Asc.

Response: We performed the control experiments as proposed by the reviewer. Specifically, we preincubated for 15 min at 37 °C two reactions: 1) CuSO₄ (0.05 mM) with α₂WD (0.05 mM) and 2) Cu^{II}WD (0.05 mM). After preincubation, HEWL (0.02 mM) and Asc (1 mM) were added. As can be seen in the **Supplementary Fig. 10** of the revised Supplementary Information, the results are consistent with our previous conclusions, as they showed the low cleavage activity of the mixture CuSO₄/ α₂WD in comparison to Cu^{II}WD. To complement our discussion, we have mentioned these new control experiments in the paragraph about ‘Control experiments’ in the section ‘POM ligand enhances Cu reactivity and selectivity’.

7. lines 200-202: “These results evidence the POM role extends beyond being a simple ligand with a greater affinity for the protein surface, it actually resembles the nature of enzymes, whose reactivity is enabled only after an enzyme substrate complex is formed in solution.” In my opinion, both parts of the sentence have very similar meanings, but if I understand well the difference, I don’t think the authors have really that evidence. Maybe a control experiment with high ionic strength medium to screen ionic interactions could provide insight into that. That would also help defining the nature of the recognition interactions beyond the crystal structure.

Response: We thank the reviewer for this nice suggestion. We have performed the oxidative cleavage reaction under two conditions of high ionic strength. First, we used 10 mM tris-HCl buffer pH 7.5 and 0.5 mM NaCl, but no fragments were observed in the SDS-PAGE. However, the high ionic strength solution caused partial precipitation of HEWL. Therefore, we conducted the reaction using a higher concentration of the Tris buffer solution (100 mM tris-HCl pH 7.5), as the Tris buffer has been previously reported to act as a shield around POMs (*Inorg. Chem.* 2021, 60, 12671). The amount of cleavage observed by SDS-PAGE was negligible (see **Supplementary Fig. 13** in the revised supporting information) in comparison to the cleavage activity at 10 mM tris-HCl pH 7.5. These results are in agreement with the decreased affinity of POMs towards protein in high ionic strength medium previously reported (*Frontiers in Molecular Biosciences* 2022, 9, 906390). The high concentration of ions in solution likely disrupts the H-bonding, and polar interactions between the POM ligand negative shell, and the protein surface, which are most likely the ones responsible for the enzyme-like recognition we observed in this work – see also reference 34 of the revised manuscript for an overview of the previously documented POM-protein interactions under similar conditions (*Acc Chem Res* 2021, 54, 1673).

8. Line 248: which are the four narrow regions for the modification? In the context of oxidative cleavage and binding it was limited to three regions. Could the authors clarify that?

Response: The oxidative cleavage was found to be in the vicinity of the three reported binding sites. Binding sites 1 (B1) consists of two different regions of the protein sequence as can be seen from Figure S27. Therefore, in addition to binding sites B2 and B3, we can see that four narrow regions of the protein sequence were susceptible to the oxidative cleavage.

9. Concerning other techniques. In the supporting information CD results are discussed. The authors report minima “at $\lambda = 208$ nm and a smaller minimum at $\lambda = 222$ nm, both characteristic for α -helical structure elements. The minimum at $\lambda = 215$ nm, characteristic of β -sheet elements...” I am not able to distinguish the three minima in Figure S16, specially the one at 215 nm. It would be helpful if the authors indicate them in the graph.

Response: We thank the reviewer for this suggestion. As suggested, we now pointed in the CD spectrum in **Supplementary Fig. 21** the position of the minima at $\lambda = 208$ and 222 nm, which are more clearly visible for HEWL. HEWL contains only a minor β -strand region, and high concentrations of HEWL are needed for the minimum at $\lambda = 215$ nm to be clearly seen. To make this point clear, we have added the following references to the CD section in the Supplementary Information: *J Agric Food Res* 2019, 1, 100004, and *J. Biol. Chem.* 1975, 250, 6977.

10. Finally, I have detected these typos and mistakes in references to figures:

L84: Asc abbreviation for ascorbate has not been introduced yet.

Response: We have introduced the Asc abbreviation in the abstract, and in line 82 of the revised manuscript.

L141: It should be contrast instead of contrasts.

Response: We have replaced contrasts by contrast as suggested.

L212 and L221: I think it should be Figure S21 in place of S19. Figure 4 would be even better as Figure S21 is already included in Figure 4.

Response: We have revised this passage and corrected the referred figure numbering.

L224: Figure S18 is an NMR spectrum. Maybe Figure S21 or Figure 4 is more adequate here.

Response: We have replaced Figure S18 with the correct reference for Figure 4.

L245: it should be copper-mediated instead of copper-mediate.

Response: We have revised the text as suggested.

11. Possible mistakes in the supporting information:

a) In the section “Preparation of known compounds”, it should be α instead of “a” in several compounds.

Response: We have replaced “a” with “ α ” as suggested.

b) In the circular dichroism section there are several places where it should be λ instead of l.

Response: We have thoroughly revised this section and replaced “l” by “ λ ” as suggested.

c) Table S1, entry 79: should it be purple?

Response: We appreciate reviewer’s careful reading of our manuscript. As correctly pointed, we have revised Entry 79 color from green to purple.

B. Non-scientific changes

To comply with the instructions provided by email, the following major non-scientific changes were done to the main text, and to the supplementary information files

1. Figure 1 has been changed from bar graph to plot style to comply with the editorial request “*Please replace your bar graphs with plots that feature information about the distribution of the underlying data*”

2. A short title has been added to all figures.

REVIEWERS' COMMENTS

Reviewer #2 (Remarks to the Author):

The authors were very positive regarding the comments. The paper was clearly improved.

Reviewer #4 (Remarks to the Author):

The authors have provided a revised manuscript in which all main points raised by the referees have been addressed in full.

Specifically, the revised manuscript is stronger in terms of mechanistic understanding of the system provided, it is clearer in terms of discussion of the observed phenomena and it is fully up to date regarding the literature. In addition, the authors have now provided a full suite of reference and comparison experiments which complement the initial manuscript and provide a clearer picture of the unique performance of the system reported.

In sum, the manuscript can now be recommended for publication without any further changes required.

Reviewer #5 (Remarks to the Author):

The authors have successfully addressed the requested additions and corrections, therefore, publication is recommended.